# Effect of Polymer Substrate on Adhesion of Electroless Plating in Irradiation-Based Direct Immobilization of Pd Nanoparticles Catalyst

**DOI:** 10.3390/nano12224106

**Published:** 2022-11-21

**Authors:** Naoto Uegaki, Satoshi Seino, Yuji Ohkubo, Takashi Nakagawa

**Affiliations:** Graduate School of Engineering, Osaka University, 2-1, Yamada-oka, Suita-shi, Osaka 565-0871, Japan

**Keywords:** radiation, Pd nanoparticles, modification, adhesion strength, Cu plating

## Abstract

Pd nanoparticles were directly immobilized on acrylonitrile–butadiene–styrene copolymer (ABS), acrylonitrile–styrene copolymer (AS), polystyrene (PS), polyphenylene sulfide (PPS), poly(vinyl chloride) (PVC), polypropylene (PP), and polyethylene (PE) polymer substrates via chemical reactions induced by ionizing irradiation. X-ray photoelectron spectroscopy analysis revealed that the chemical state of the immobilized Pd nanoparticles depended on the polymer substrate type. Electroless plating was performed using the immobilized Pd nanoparticles as the catalyst, and Cu-plating films were deposited on all polymer substrates. The results of the tape-peeling test suggested that the chemical state of the immobilized Pd nanoparticles on the polymer substrates affected the plating adhesion strength. Notably, ABS with immobilized Pd particles exhibited a high adhesion strength beyond the practical level, even without prior chemical etching. It was presumed that the high adhesion strength was owing to the anchoring effect of the holes generated on the ABS surface by ionizing irradiation.

## 1. Introduction

Metallized plastics are widely used in various fields, such as automotive, home appliance, and water-related component manufacturing. Metallization imparts the treated materials with reflectivity, abrasion resistance, electrical conductivity, and decorative properties. Electroless plating is generally used for metal coating on nonconductive materials such as plastics. Acrylonitrile–butadiene–styrene copolymer (ABS) is currently the most widely used polymer substrate for plating, owing to its mass producibility, dimensional stability, impact resistance, and high processability. For electroless plating on ABS substrates, chemical etching using a chromic acid–sulfuric acid solution is typically performed for pretreatment. The applied chromic acid dissolves the polybutadiene phase in the treated ABS substrate, which increases its surface roughness and significantly improves mechanical adhesion [1,2,3,4]. Other polymer substrates are difficult to etch, making it difficult to adhere the Pd catalyst to the polymer substrate and deposit electroless plating films. Several studies have sought to overcome this problem. For example, methods that add a filler to a resin matrix or apply a strong acid together with an organic solvent have been proposed for chemical etching [5,6,7,8,9,10]. Alternatively, presurface treatment (e.g., using UV irradiation) has been proposed as a physical etching process [11,12,13].

Our research group has been working on the development of methods for direct immobilization of noble metal nanoparticles on various substrates, such as ceramics, carbon, textile fabrics, and polymer substrates, using chemical reactions induced by ionizing radiation [14,15,16]. The process of nanoparticle formation induced by ionizing radiation has been elucidated by Henglein and Belloni [17,18,19]. We have found that metal nanoparticles can be directly immobilized on the support surface by irradiating the precursor aqueous solution containing the support. Recently, we reported that Pd nanoparticles immobilized on ABS substrates worked as a catalyst for electroless plating, and the Cu plating films obtained using this method exhibited a high adhesion strength despite no etching treatment [16]. This is a novel catalyst-immobilization method that differs from the graft polymerization method, in which functional groups are introduced by irradiating the polymer surface [20].

In the present study, we investigated the applicability of the above-described method to various polymer materials. Because ABS is a copolymer consisting of three polymers (acrylonitrile, butadiene, and styrene), the components that contribute to its high adhesion were investigated using polystyrene (PS) and acrylonitrile–styrene copolymer (AS). In addition to these three substrates, the possibility of Pd immobilization on various substrates was examined using homopolymers such as polypropylene (PP), a commodity plastic, and polyphenylene sulfide (PPS), an engineering plastic. As noble metal species, we used palladium, which is the most common catalyst for electroless plating reactions.

## 2. Materials and Methods

### 2.1. Immobilization of Pd on Substrates

The chemicals used were Pd (II) nitrate (Pd(NO_3_)_2_; Tanaka Kikinzoku Kogyo K.K., Tokyo, Japan) and 2-propanol (CH_3_CH(OH)CH_3_; Wako, Osaka, Japan). Commercial ABS ((CH_2_CH(C_6_H_6_)·CH_2_CHCHCH_2_·CH_2_CH(CN))*_n_*, AS-ONE, Osaka, Japan), AS ((CH_2_CH(C_6_H_6_)·CH_2_CH(CN))*_n_*, Toray, Tokyo, Japan), PS ((CH_2_CH(C_6_H_6_))*_n_*, Hikari Co., Ltd., Osaka, Japan), PPS ((S(C_6_H_4_))*_n_*, DIC Corporation, Tokyo, Japan), poly(vinyl chloride) (PVC) ((CH_2_CHCl)*_n_*, AS-ONE, Osaka, Japan), PP ((CH_2_CH(CH_3_))*_n_*, AS-ONE, Osaka, Japan), and polyethylene (PE) ((CH_2_CH_2_)*_n_*, AS-ONE, Osaka, Japan) were used as substrates. Before use, the substrates were washed with 2-propanol and pure water, and then dried at room temperature. Plastic zipper packs (Uni-Pack E-4, Seisannipponsha Ltd., Tokyo, Japan) were used as the reaction vessels.

An aqueous solution of Pd(NO_3_)_2_ (1 mM) was used as the precursor. The substrates were immersed into 20 mL of the precursor solution. Plastic zipper packs containing the substrates and precursor solutions were lined on the irradiation tray. The samples were irradiated with a high-energy electron beam (4.8 MeV) using a dynamitron accelerator (SHI-ATEX; Tokyo, Japan). The surface dose was 20 kGy. These irradiation conditions are comparable to sterilization of medical devices. It was assumed that electron-beam irradiation induced chemical reactions that generated Pd nanoparticles and simultaneously introduced functional groups on the surface of the polymer substrate. Pd nanoparticles formed by the irradiation were expected to be immobilized on the polymer substrate surface [11]. After the irradiation, the substrates were removed from the solution and washed with pure water using an ultrasonic cleaner for 5 min. The washed samples were then dried at room temperature. The obtained samples were denoted as Pd/ABS, Pd/AS, Pd/PS, Pd/PPS, Pd/PVC, Pd/PP, and Pd/PE.

### 2.2. Material Characterization

All the obtained samples were characterized by scanning electron microscopy (SEM), inductively coupled plasma atomic emission spectroscopy (ICP-AES), and X-ray photoelectron spectroscopy (XPS). The surfaces of the irradiated samples were observed by SEM-EDX (JSM-7001F, JEOL, Tokyo, Japan). Prior to the SEM observations, the surfaces of the polymer plates were coated with Os (OPC60A, Filgen, Aichi, Japan). SEM observations were performed at an acceleration voltage of 10–15 kV. The amount of Pd immobilized on the substrates was analyzed by ICP-AES (ICPE-9000, Shimadzu, Kyoto, Japan). The substrate surfaces (50 mm × 50 mm) were treated with 2 mL of aqua regia (HCl:HNO_3_ = 3:1) to dissolve the Pd nanoparticles. The surface chemical state was analyzed by XPS (Quantum 2000, ULVAC-PHI, Kanagawa, Japan) using an Al *K*α X-ray source operating at 15 kV. A C1*s* level of 284.6 eV was used as an internal standard to correct for the peak drift. All high-resolution XPS peaks were fitted using XPSPEAK41 software.

### 2.3. Electroless Plating Process

The electroless plating process described in this study consisted of two main steps:

(i)Acceleration step

In this step, the chemical states of immobilized Pd were converted to metallic sate to increase their activity. Before acceleration, the wettability was imparted by dipping the substrates in a 5.0 vol% Thru-Cup MTE-1-A (C. Uyemura & Co., Ltd., Osaka, Japan) aqueous solution at 50 °C for 2 min. After the pretreatment, the samples were washed with water at 50 °C for 1 min and rinsed with water at room temperature. The accelerators used were ALCUP Reducer MAB-4-A, MAB-4-C, and MRD-2-C (C. Uyemura & Co., Ltd., Osaka, Japan). The Pd-immobilized samples were accelerated using the following composition: 1.0 vol% MAB-4-A, 5.0 vol% MAB-4-C, and 1.0 vol% MRD-2-C in an aqueous solution.

(ii)Electroless Cu plating step

The surface-activated Pd/ABS substrates were placed in electroless Cu plating baths. The conditions of the aqueous solution in the plating bath were as follows: 10 vol% Thru-Cup PEA-40-M, 6.0 vol% Thru-Cup PEA-40-B, 3.5 vol% Thru-Cup PEA-40-D, and 2.3 vol% 37% formaldehyde. The temperature of the plating bath was maintained at 34 °C. The electroless Cu plated ABS substrates were then rinsed with distilled water and dried under warm air. Subsequently, they were subjected to a thermal treatment at 80 °C for 60 min.

### 2.4. Electroplating Process

Electroplating was necessary for measuring the adhesion strength of electroless Cu plating on substrates in the 90° peel test. Acid degreasing and acid activity treatment were performed as pretreatments. For acid degreasing, the substrates were immersed into 10 vol. % DP-320 (Okuno Chemical Industries Co., Ltd., Osaka, Japan) aqueous solution at 40 °C for 5 min. For the acid activity treatment, the substrates were immersed into 1.82 mol/L H_2_SO_4_ at room temperature for 30 s. Electroplating was carried out in a 0.5 vol% Top Lucina 2000MU (Okuno Chemical Industries Co., Ltd., Osaka, Japan), 0.05vol% Top Lucina 2000A (Okuno Chemical Industries Co., Ltd., Osaka, Japan), 1.25 mol/L CuSO_4_, 0.5 mol/L H_2_SO_4_, and 0.0175 vol% conc. HCl solution at 25 °C. The cathode current density was maintained at 3 A/dm² for 1 h.

### 2.5. Tape-Peeling Test

The adhesion of the electroless Cu plating was evaluated using the tape-peeling test based on JIS H 8504:15, 1999. A strip of the CT405AP-18 cellophane tape (Nichiban Co., Ltd., Tokyo, Japan) was applied to less than 8 mm of the substrate and adhered to the plating film by rubbing three times with a finger. The tape was then peeled off such that it was perpendicular to the plating surface.

### 2.6. Peel Adhesion Strength Test

The adhesion strength was determined by measuring the 90° peel adhesion strength using AG-Xplus (Shimadzu, Kyoto, Japan), with a 1-kN-load cell. All measurements were carried out at a width of 10 mm, peel-off speed of 50.0 mm/min, and constant temperature of 24 °C. The adhesion strength was reported as the average of three sample measurements.

## 3. Results

### 3.1. Characterization of Pd/Substrates

Figure 1a–g show the SEM images of the Pd/substrate samples. The SEM-EDX analysis revealed that the small nanoparticles, manifesting as white spots, consisted of Pd. Pd nanoparticles with diameters in the 20–30 nm range were observed, together with relatively large aggregates with 80–500 nm diameters, in all samples (Appendix A). The amount of Pd nanoparticles immobilized on the polymer substrates were 2.4, 1.9, 1.2, 1.8, 1.5, 5.2, and 6.2 μg/cm^2^, for Pd/ABS, Pd/AS, Pd/PS, Pd/PPS, Pd/PVC, Pd/PP, and Pd/PE, respectively. These results clearly indicate that Pd nanoparticles were successfully immobilized on the surfaces of the studied polymer substrates.

Figure 2 and Figure 3 show the XPS analysis results for the Pd/substrate samples. Of note, the shape of the Pd3*d* XPS spectrum depended on the polymer substrate (Figure 2). The XPS peaks of Pd3*d* for Pd/PS, Pd/PPS, Pd/PP, and Pd/PE were 335.2 eV and 340.5 eV, corresponding to the metallic Pd [21]. For Pd/PVC, peaks corresponding to the metallic state and PdCl_2_ (338 eV and 343 eV) [22] were observed. In contrast, the peaks obtained for Pd/ABS were 339.2 eV and 344.5 eV (Figure 2). These peaks were attributed to the state in which Pd was coordinated with the oxygen atoms in the carbonyl groups [23]. For Pd/AS, peaks corresponding to the metallic state and coordination of carbonyl groups with oxygen were observed.

The chemical states of the polymer substrates before and after the Pd particle immobilization were evaluated using XPS O1*s* and C1*s*. After the irradiation, the peak intensities of O1*s* increased for Pd/ABS, Pd/AS, Pd/PVC, Pd/PP, and Pd/PE (Figure 3a). As the polymers constituting these substrates did not contain oxygen, it was assumed that oxygen-containing functional groups were generated on the polymers’ surfaces owing to the electron beam irradiation [24]. Notably, the XPS spectra of Pd/ABS and Pd/AS exhibited a peak at approximately 535.9 eV, which was quite close to the value reported for Pd(acac)_2_ [25]. This result suggests that Pd coordinated with the carboxyl groups formed on the ABS and AS surfaces. In contrast, the O1s and C1s spectra of Pd/PS and Pd/PPS were almost unchanged before and after the Pd particle immobilization by irradiation.

### 3.2. Electroless Cu Plating of Pd/Substrates

Figure 4a–g show exterior images of the samples after the electroless plating step. These images clearly suggest that Cu plating films were successfully obtained for all the samples. Thus, Pd nanoparticles immobilized on the polymer substrates functioned as plating catalysts, regardless of their chemical states. The surface coverage of the Cu plating films depended on the polymer substrate. The surfaces of the polymer substrates were completely covered with Cu plating films for Pd/ABS, Pd/AS, Pd/PS, and Pd/PP. In contrast, region on which no copper plating occurs were observed for Pd/PPS, Pd/PVC, and Pd/PE.

The adhesion properties of the Cu plating films depended on the polymer substrate. The Cu plating films did not peel off for Pd/ABS and Pd/PPS (Figure 5a,d). For Pd/AS and Pd/PVC, the Cu plating films partially peeled off (Figure 5b,e).

For Pd/PS, Pd/PP, and Pd/PE, most of the Cu plating films peeled off at the tested locations (Figure 5c,f,g).

## 4. Discussion

The results for the Pd immobilization amount, chemical state of Pd, deposition of electroless Cu plating films, and tape peeling tests are summarized in Table 1.

### 4.1. Relationship between the Chemical State of Substrate Surface-Immobilized Pd and the Polymer Substrate Type

The chemical reactions induced by the electron-beam irradiation were of two types: (1) in-solution reduction reactions of Pd ions and (2) polymer surface modification reactions. In solution, water radiolysis proceeded to form hydrated electrons, H radicals, and OH radicals [26,27]:H_2_O ⟿ e_aq_^−^, H^•^, OH^•^, etc.(1)

The generated hydrated electrons and H radicals reduced the Pd^2+^ ions in solution to form Pd nanoparticles [18]:Pd^2+^ + 2e_aq_^−^ → Pd^0^(2)
Pd^2+^ + 2H^•^ → Pd^0^ + 2H^+^(3)

The irradiation-induced reduction of Pd^2+^ ions did not depend on the polymer substrate type. The modification of the polymer substrates by the ionizing radiation in the present study included direct modification, such as crosslinking and decomposition by radiation, and indirect modification by chemical reactions with active species generated by water radiolysis. During the direct modification by radiation, radicals were generated in polymer chains, and crosslinking and decomposition reactions proceeded. During the indirect modification, the oxidizing species generated by water radiolysis attacked the polymer chains to proceed with the oxidation reaction. The degree of the modification differed, depending on the stability with respect to the radiation and chemical stability of specific polymer substrate.

The relationship between the chemical state of the polymer substrates and the immobilized Pd nanoparticles on the substrates is discussed below. The O1*s* and C1*s* XPS spectra of Pd/PPS and Pd/PS with aromatic rings changed little before and after irradiation. Polymers with aromatic groups are known to be radiation-stable because they efficiently dissipate energy [26,27]. Because these substrates were radiation-stable, functional groups were not introduced to the substrate surface. Therefore, Pd was assumed to be immobilized in the metallic state. The O1*s* and C1*s* XPS spectra of Pd/ABS, Pd/AS, Pd/PVC, Pd/PP, and Pd/PE exhibited changes before and after irradiation. It was confirmed that functional groups were introduced onto the surfaces of these substrates by irradiation. For Pd/PP and Pd/PE, functional groups were formed on the substrate surface upon irradiation, but the chemical state of the immobilized Pd was metallic. In contrast, for Pd/AS, the chemical state of Pd was metallic and coordinated to the oxygen of the carbonyl group. For Pd/ABS, the chemical state of Pd was coordinated to the oxygen of the carbonyl group. Because XPS is a surface-sensitive analytical method, only the surfaces of Pd nanoparticles can be interpreted as coordinating to the carbonyl group. Exceptionally, for Pd/PVC, the chemical state of Pd was PdCl_2_-like in addition to metallic. Halogens have very high electron affinity. Halides cause dissociative electron addition reactions via reactions with secondary and other electrons in the system. The dissociated Cl^−^ ions are considered to be coordinated to Pd. The radiation sensitivity of the polymer substrate strongly affected the chemical state of the immobilized Pd nanoparticles.

### 4.2. Properties of Cu Plating Films and Chemical State of Pd

#### 4.2.1. The Surface Coverage of Cu Plating Films

In the electroless Cu plating reaction, Pd immobilized on the substrate functioned as a catalyst to promote the formaldehyde oxidation reaction and generate electrons. The generated electrons caused the Cu^2+^ ion reduction reaction to proceed by depositing Cu plating films. As shown in Figure 3, electroless Cu plating films were deposited on all samples. Pd nanoparticles immobilized on the polymer substrates using this method worked as electroless Cu-plating catalysts. As previously mentioned, all samples were treated with an accelerant solution prior to the electroless plating process. The Pd species immobilized on the substrates were reduced by the acceleration step to form metallic Pd, which exhibited catalytic activity. On samples where Pd was immobilized in the metallic state, electroless Cu-plating films were deposited without the acceleration step (Appendix A). The deposition rates of the Cu plating films on different polymer substrates are discussed below. Electroless plating films were deposited on the entire surfaces of the ABS, AS, PS, and PP samples. However, the Cu plating films were not deposited in some areas for the PPS, PVC, and PE samples. Because there was no correlation with the loading amount of Pd, it was assumed that the plating coverage depended on the distribution of Pd on the polymer surface. These results suggest that the immobilization process of Pd nanoparticles should be improved for some polymer substrates.

#### 4.2.2. Adhesion Strengths of Cu Plating Films

The adhesion characteristics of electroless Cu plating films depended on the Pd-immobilized polymer substrate type. The plating films for PS, PP, and PE were easily peeled off, and Pd nanoparticles were immobilized in the metallic state. In contrast, the plating films for AS and PVC exhibited a relatively high adhesion, in which Pd existed in a metallic state and/or coordinated to functional groups on the substrates. In particular, the plating film for ABS did not delaminate at all, in which Pd was coordinated to the functional groups only. These results suggest that the coordination of Pd to the functional groups on the polymer surfaces is one of the factors responsible for their high plating-adhesion strength. As an exception, the plating film for the PPS surface did not peel off even though Pd nanoparticles were immobilized in the metallic state; this should be discussed in further studies.

Let us consider the reason why the Pd/ABS samples exhibited particularly high adhesion. Electrolytic plating was performed for Pd/ABS and Pd/AS covered with electroless Cu plating, to measure the peeling strength. The peeling strength was 0.03 N/mm for Pd/AS. On the other hand, the peeling strength for Pd/ABS was 0.82 N/mm, exceeding the minimum permissible adhesion strength of 0.7 N/mm specified in the printed circuit standard. The surfaces of the ABS and AS substrates after the peeling-strength test were observed using SEM, and the results are shown in Figure 6. The surface of the ABS substrate was significantly roughened after the Cu plating was peeled, suggesting that the cohesive failure of the ABS substrate occurred during the peeling process. On the other hand, the surface of the AS substrate did not show any unevenness after the removal of the Cu plating, indicating that the cohesive failure of the AS substrate did not occur. The significantly higher peeling strength for Pd/ABS suggests the existence of mechanical anchoring, even without etching. For further discussion, the surfaces of ABS and AS substrates before and after irradiation were observed with SEM, and the images are shown in Appendix A. For ABS samples, some holes with diameters of approximately 80 nm were observed on the irradiated ABS surface, while no holes were observed for AS (Appendix A). From these results, it can be inferred that the decomposition of butadiene during irradiation might lead to the formation of small holes, which contributed to the mechanical anchoring of the Cu plating films.

## 5. Conclusions

Pd was immobilized on all the polymer substrates using a radiation-based method. Electroless Cu plating was performed for all the polymer substrates using immobilized Pd. The adhesion characteristics of the deposited electroless Cu plating films depended on the substrate type. It was suggested that the immobilization of Pd on the polymer substrate via its functional groups contributed to this. For Pd/PS, Pd/PE, and Pd/PP, in which Pd was immobilized in the metallic state, the adhesion of the plated films was low. In contrast, the adhesion of the plating films was high for Pd/ABS, Pd/AS, and Pd/PVC, where the chemical state of the immobilized Pd was coordinated to the functional groups. Notably, Pd/ABS exhibited strong adhesion and high peeling strengths, beyond the practical level. It was presumed that the high adhesion was owing to the anchoring effect of the holes formed on the ABS substrate surface by the radiation-induced decomposition of the butadiene components. The direct immobilization of noble metal nanoparticles onto polymer substrates is a unique and unparalleled technique. This technique is expected to be applicable to other polymer substrate types and noble metal species.

## Figures and Tables

**Figure 1 nanomaterials-12-04106-f001:**
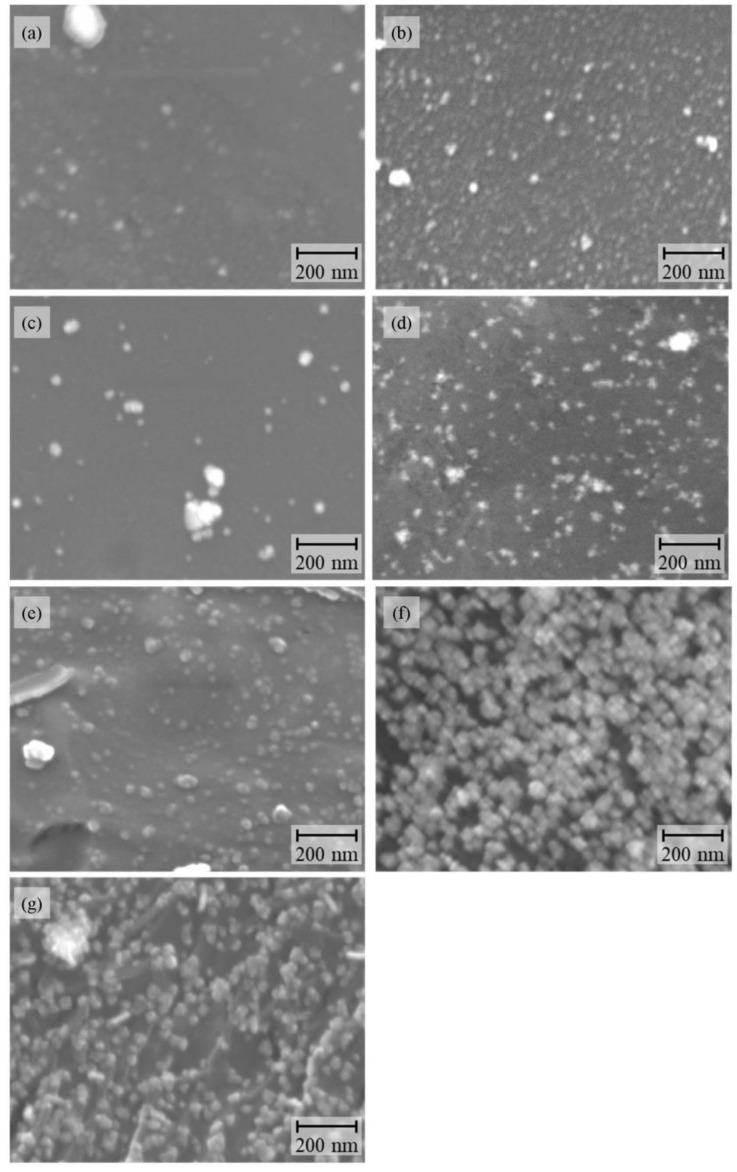
SEM images of Pd/substrates: (**a**) Pd/ABS, (**b**) Pd/AS, (**c**) Pd/PS, (**d**) Pd/PPS, (**e**) Pd/PVC, (**f** )Pd/PP, and (**g**) Pd/PE.

**Figure 2 nanomaterials-12-04106-f002:**
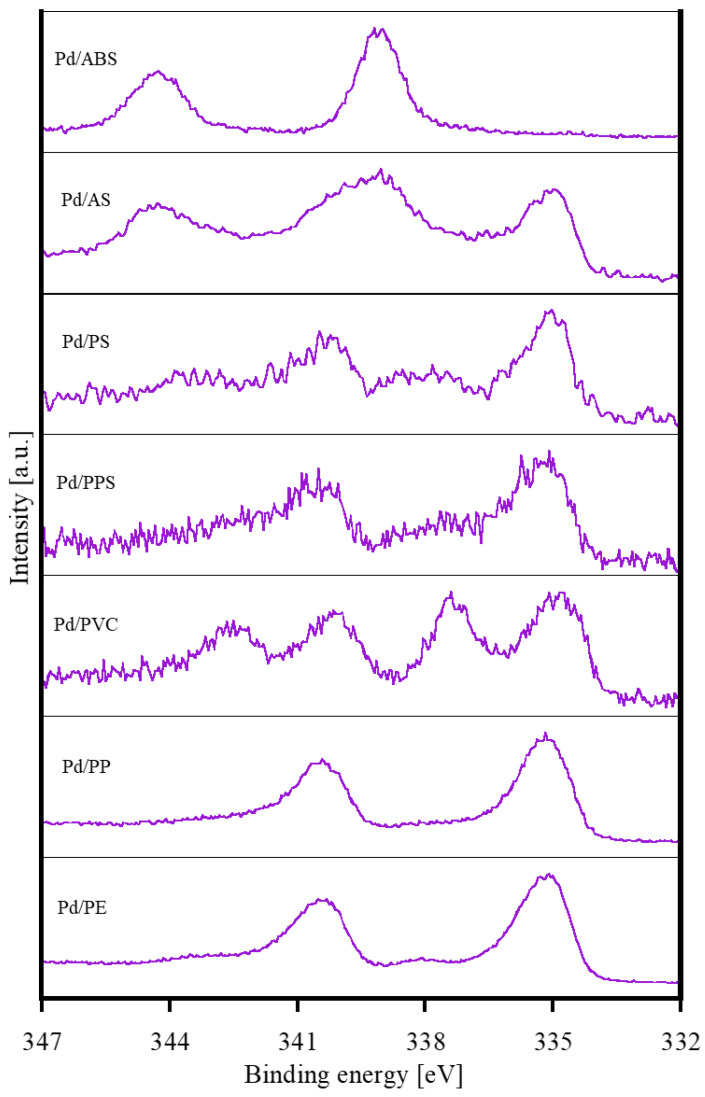
Pd3*d*-XPS spectra, for different Pd/substrate scenarios. In each case, the XPS spectra were normalized by the highest peak intensity.

**Figure 3 nanomaterials-12-04106-f003:**
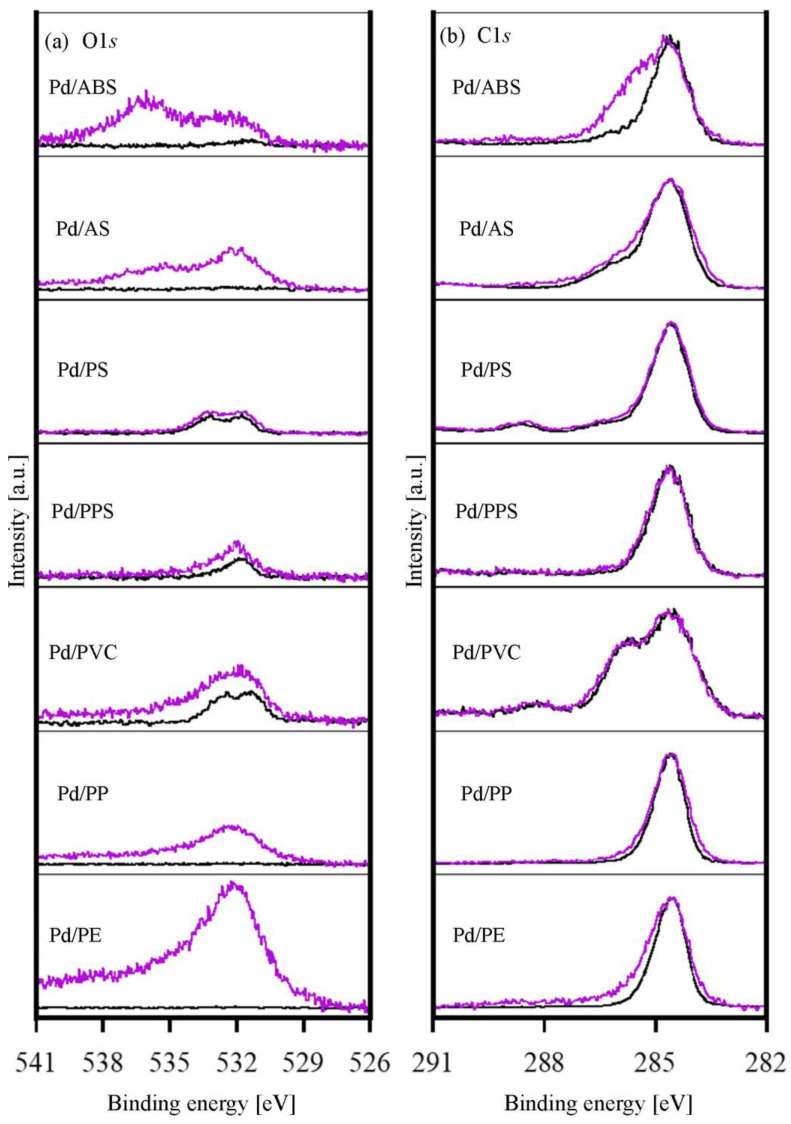
XPS spectra normalized to the peak intensity at 286.4 eV. (a) O1*s*, (b) C1*s*. The purple lines are the spectra after the Pd particle immobilization on the polymer substrates. The black lines are the spectra of as-received substrates.

**Figure 4 nanomaterials-12-04106-f004:**
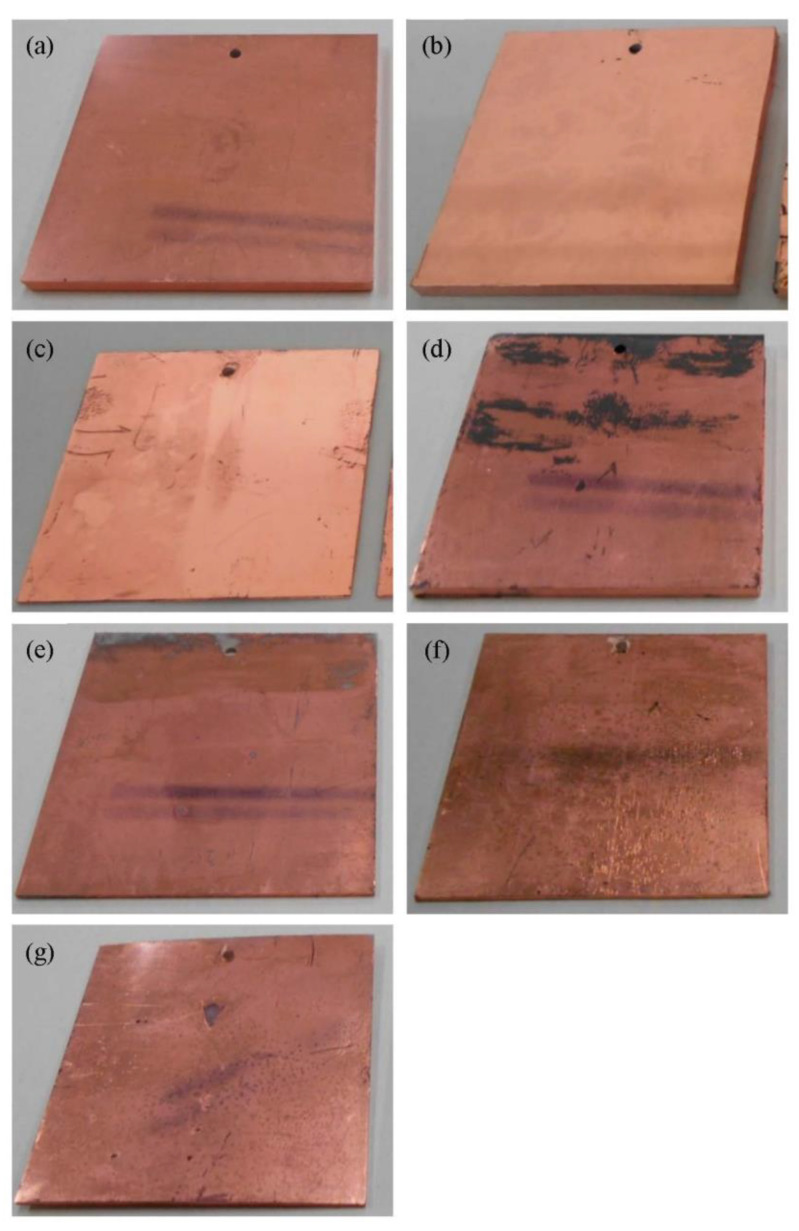
Exterior pictures of Pd/substrate samples after electroless Cu plating: (**a**) Pd/ABS, (**b**) Pd/AS, (**c**) Pd/PS, (**d**) Pd/PPS, (**e**) Pd/PVC, (**f**) Pd/PP, and (**g**) Pd/PE.

**Figure 5 nanomaterials-12-04106-f005:**
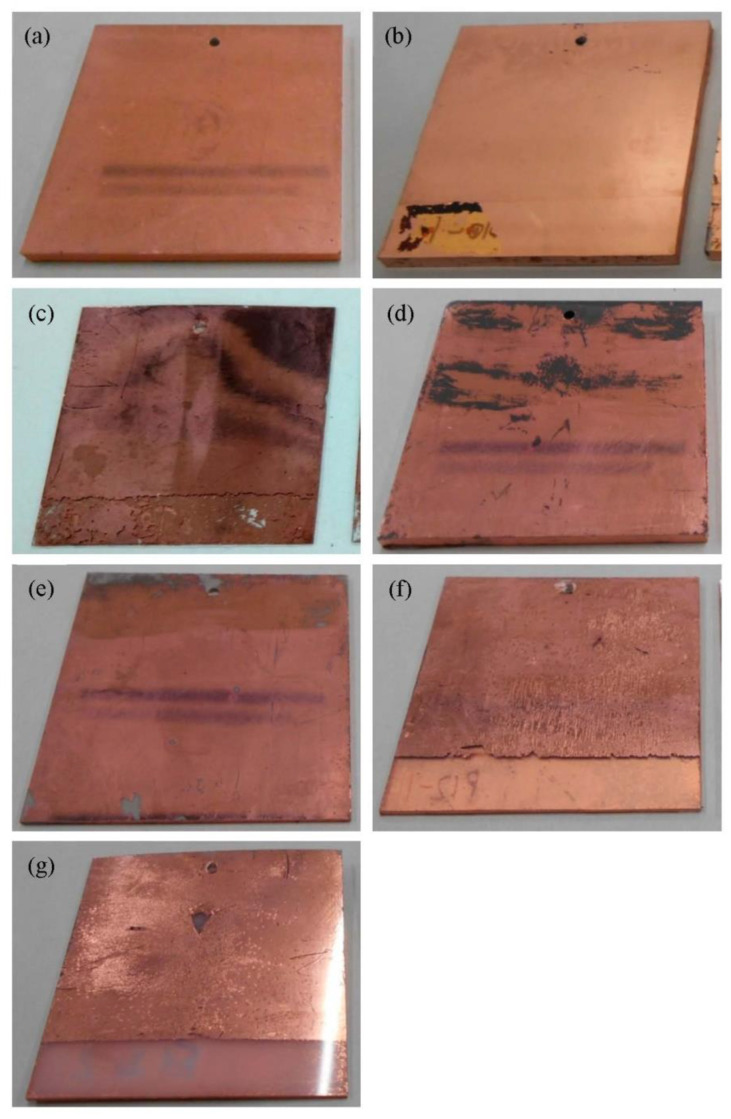
Exterior pictures of Pd/substrate samples after the tape-peeling test: (**a**) Pd/ABS, (**b**) Pd/AS, (**c**) Pd/PS, (**d**) Pd/PPS, (**e**) Pd/PVC, (**f**) Pd/PP, and (**g**) Pd/PE. The tape-peeling test was performed at the lower 8 mm of the samples.

**Figure 6 nanomaterials-12-04106-f006:**
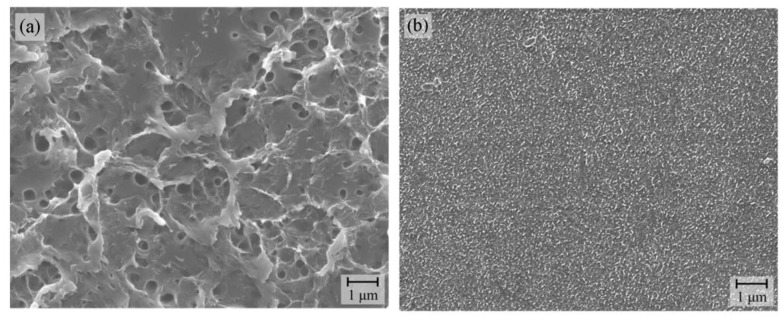
SEM images of substrates after peel electroless Cu plating. (**a**) Pd/ABS, (**b**) Pd/AS.

**Table 1 nanomaterials-12-04106-t001:** Amounts and chemical states of immobilized Pd, for different polymer substrates, and the deposition and adhesion of electroless Cu plating films.

Substrate	Pd Amount [µg/cm^2^]	Pd chemical State (XPS Pd3*d*)	Plating Coverage (Visual Confirmation)	Adhesion (Tape-Peeling Test)
ABS	2.4	Functional group coordination	Fully deposited	No peeling
AS	1.9	Metal, Functional group coordination	Fully deposited	Partial peeling off
PS	1.2	Metal	Fully deposited	Peeling off
PPS	1.8	Metal	Deposition failure	No peeling
PVC	1.5	Metal, PdCl_2_ like	Deposition failure	Partial peeling off
PP	5.2	Metal	Fully deposited	Peeling off
PE	6.2	Metal	Deposition failure	Peeling off

## Data Availability

The data presented in this study are available upon request from the corresponding author.

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
