# Peer review of "Effect of Polymer Substrate on Adhesion of Electroless Plating in Irradiation-Based Direct Immobilization of Pd Nanoparticles Catalyst"

_nanomaterials, 2022, doi:10.3390/nano12224106_

Round 1

Reviewer 1 Report

The manuscript presents original, scientifically, and technologically valuable results on the effect of polymer substrate on the adhesion of electroless plating in irradiation-based direct immobilization of Pd nanoparticles catalyst. The performed experiments suggested that the chemical state of the immobilized Pd nanoparticles on the polymer substrates affected the plating adhesion strength. Additionally, Pd/ABS exhibited strong adhesion and high peeling strengths, beyond the practical level. These are valuable technological messages. The manuscript is well written, easy to follow, and understand the main ideas.  

Despite all benefits, it would be worthy to mention the practical applicability of received results in conclusions as the use of Pd and irradiation with a high-energy electron beam (4.8 MeV; surface dose - 20 kGy) could make this technological method rather expensive.

Reviewer 2 Report

The authors have presented a systematic study, showing the importance of understanding the various factors affecting the adhesion of plated Cu films on nanoparticle-activated polymer surfaces. I recommend publication of the work, however, a few questions / remarks should be taken into account in a revised version of the manuscript, all of which are listed below: 

Line 30

[ productivity] > is that the word with the correct meaning in this context ?

[producibility] ?

Line 94

[In this step, the immobilized Pd particles were converted to Pd0] > what was the original chemical oxidation state of the ‘Pd particles’ ? In case that are Pd2+ ions, these should not be called ‘particles’, as that can lead to confusion.

Line 94: why is it called ‘acceleration step’ ? Acceleration of what exactly ? Isn’t ‘activation step’ a more correct formulation ?

Line 96-101

Many commercial product names are mentioned in that section, e.g. [Thru-Cup MTE-1-A]. Can the authors add some comments on what these products contain as there active ingredient ?

Line 104-105: same comment as previous one.

Line 114: same comment

Line 116-117: same comment

Line 171

[undeposited Cu-plating film areas] > this is a very complex and confusing way of formulating a region on which no copper plating occurs ! Please reformulate.

Table 1

[peeling off] should be [Peeling off]

Table 1

There are 4 entries [Metal,] that should be [Metal]

Reaction 1 in between line 186 and 187: the [etc] is intriguing ! The authors should add a reference, to indicate the complexity of the radiolysis processes that can occur, so the reader can refer to it.

Line 187 + 189

[Pd ions] > [Pd2+ ions]

Line 221

[The plating coverage of Cu plating films] > [The surface coverage of Cu plating films]

Line 224

[Cu ion] > [Cu2+ ion]

Line 230

[In samples] > [On samples]

Line 256 + 280

What do the authors mean by [practical level of 0.7 N/mm] ? That should be properly explained. Is this a requirement for applications ?

Line 273

[for both polymer substrates] > Which 2 substrates do the authors mean ? Please reformulate. 

Reviewer 3 Report

The atuhors of the manuscript "Effect of polymer substrate on adhesion of electroless plating in irradiation-based direct immobilization of Pd nanoparticles catalyst" aimed to immobilize Pd NPs on several types of (co)polymers by ionizing radiation induced chemical reactions.

The manuscript is very interesting, however, before publication, the authors need to address the following aspects:

1. Why did the authors choose to explore PdNPs immobilization? What is the practical application envisaged? These aspects need clarification both in the Introduction section and in the Abstract

2. The synthesis of metal nanoparticles using ionizing radiation represents the subject of multiple works, some of which should be cited in the introduction chapter

3. The analytical procedures applied needs to be more exhaustively presented. Only XPS has a good presentation

4. Conclusions - what are the main findings of the study? What are the probable applications of the materials? A clearer take-home message is needed

5. The reference list should be enhanced with more recent studies

Reviewer 4 Report

Ref.comments to the paper titled as “Effect of polymer substrate on adhesion of electroless plating in irradiation-based direct immobilization of Pd nanoparticles catalyst” written by the authors: Naoto Uegaki, Satoshi Seino, Yuji Ohkubo, and Takashi Nakagawa.

It is known that currently, study of the features of the nanoparticles, using different synthesis approaches, different substrates, different geometric constructions, etc. is very useful for the knowledge and human activity, as well as for the possibility to extend the optoelectronics materials database.  From this point of view the manuscript is actual and modern.

For the first, it is remarked that the author has made short literature search, analyzing only 19 references. Moreover, the papers written by the last 5 years are analyzed not in good conditions; the authors are considered so little part of the papers written by the last 3-5 years.  Please add approximately 5-7 papers written on 2020-2022 years in the same directions.  This will indicate the knowledge of the problem, its useful application and finding ways to solve it.

Materials and Methods sections are explained well.

Results and Discussions sections. Good experimental data are shown in Figure 1: SEM images of Pd/substrates… and Figure 2: Pd3d-XPS spectra, for different Pd/substrate scenarios…,which have been  obtained using the different substrates and have shown the dependences of the properties of the Pd nanoparticles on features of the substrates. The different Pd nanopraticles properties have been supported by the results of Figure 3. XPS spectra normalized to the peak intensity at 286.4 eV. Would you please to discuss the same or other Pd nanoparticles parameters change using other range of the irradiations? It can extend our basic physical-chemical knowledge.

Subsection: Electroless Cu plating of Pd/substrates – can be useful for the engineers in their practical jobs.

Table 1: Amounts and chemical states of immobilized Pd, for different polymer substrates, and the deposition and adhesion of electroless Cu plating films – accumulates the obtained results regarded to improve the adhesion process. Equations (1)-(3) have supported the process.

Other discussion paragraphs are coincided with our physical consideration and knofledge.

Would you please to show the microhardness and laser strength of your substrate after the Pd nanoparticles placement?

Conclusion part should be extended.

So, the paper is interesting for the specific area for the researchers and students. I can recommend to the authors to answer the questions mentioned above.  Thus, the paper can be published after minor corrections.

Round 2

Reviewer 3 Report

The authors addressed all the points raised in the first round of peer-review. The manuscript can be accepted in the present form.